# Association of atherogenic index of plasma and triglyceride-glucose index with atrial fibrillation: A retrospective study

**Xuebo Li, Shangzhi Shu, Chaoqun Huang, Shuyan Li** *

Department of Cardiology, The First Hospital of Jilin University, Jilin Province, China

* li_sy@jlu.edu.cn

## Abstract

### Background

The triglyceride-glucose (TyG) index is an indirect marker of insulin resistance used to assess diabetes mellitus and cardiovascular disease risk. However, its clinical evidence regarding atrial fibrillation (AF) remains limited. Similarly, the atherogenic index of plasma (AIP), a recently discovered cardiovascular risk biomarker, has not been evaluated in relation to AF. Therefore, this study aimed to analyze the relationship between the TyG index, AIP, and AF, and compare their ability to predict AF risk.

### Methods

This retrospective study included 1,122 patients hospitalized at the First Hospital of Jilin University between January 1, 2023, and February 29, 2024. The associations between the TyG index, AIP, and AF risk were analyzed using multivariate logistic regression, stratified subgroup analyses, and restricted cubic spline regression. Correlation and mediation analyses were performed to evaluate the relationship between the two biomarkers and potential indirect interactions linking the TyG index, AIP, and AF occurrence. Receiver operating characteristic (ROC) curves were generated to compare the predictive accuracies of the TyG index and AIP for AF risk.

### Results

The TyG index and AIP were identified as independent predictors of AF development. Significant positive and nonlinear relationships were identified between both indices and AF (overall, $P < 0.001$; nonlinearity, $P < 0.001$). Subgroup analyses confirmed an elevated AF risk associated with increased TyG and AIP values across various patient subcategories, without significant interaction effects. A strong positive correlation was observed between the TyG index and AIP. Mediation analysis indicated no significant indirect effects of the TyG index and AIP on AF risk ($P = 0.132$). ROC curve analysis showed that AIP and TyG had comparable predictive abilities.

**Data availability statement:** All relevant data are within the manuscript and its Supporting Information files.

**Funding:** Initials of the authors who received each award: SL Grant numbers awarded to each author: No. 82070524 The full name of each funder: This study was supported by the National Natural Science Foundation of China. Role of Funder: The funder had no role in the conceptualization, design, data collection, analysis, decision to publish, or preparation of the manuscript.

**Competing interests:** The authors have declared that no competing interests exist.

## Conclusions

The TyG index and AIP were independently associated with increased AF risk. Additionally, AIP demonstrated predictive accuracy comparable to that of the TyG index in predicting AF risk.

## Introduction

Atrial fibrillation (AF) is the most commonly identified form of cardiac arrhythmia. With an aging population and increasing rates of obesity, diabetes mellitus (DM), hypertension, and metabolic syndrome, the global occurrence of AF is continually increasing. AF significantly increases the risk of stroke and thromboembolic events and is strongly correlated with several cardiovascular diseases, notably heart failure [1]. Consequently, timely recognition and accurate prediction of AF-associated risk factors are essential for optimizing prevention, stratifying patient risks, and enhancing clinical outcomes.

Insulin resistance (IR) significantly influences AF pathogenesis through diverse mechanisms [2]. First, IR promotes the onset of AF through structural and electrical remodeling, including left atrial enlargement, fibrosis, and intracellular calcium homeostasis imbalance [3,4]. Oxidative stress and inflammation caused by IR lead to atrial remodeling through a synergistic interaction involving mechanisms such as reactive oxygen species (ROS) damage, ion channel dysfunction, and the release of inflammatory factors [5–8]. In addition, IR can lead to autonomic nervous system disturbances, such as decreased vagal activity and dysregulated autonomic balance [9,10]. Recent studies have increasingly emphasized the advantages of the triglyceride-glucose (TyG) index over conventional IR assessment methods for accurately quantifying IR severity and predicting cardiovascular events [11]. Given the recognized contribution of IR to AF pathogenesis [2], researchers have proposed that the TyG index may serve as a meaningful marker for predicting AF onset [12]. However, clinical evidence supporting this hypothesis is limited.

The atherogenic index of plasma (AIP) is widely used to detect dyslipidemia and estimate the likelihood of atherosclerotic cardiovascular disease. Increased AIP values are strongly correlated with higher coronary artery disease (CAD) risk [13]. Moreover, AIP, recognized for its sensitivity to changes in lipid metabolism, is significantly associated with IR severity and type 2 DM development [14]. Prior research has observed significantly elevated triglyceride (TG) levels, approximately 1.6 times higher, in patients with AF, accompanied by heightened inflammatory markers [15]. Independent relationships have also been reported between reduced high-density lipoprotein cholesterol (HDL-C) levels and an elevated AF risk [16]. Given that AIP is derived from the logarithmic TG-to-HDL-C ratio, it effectively reflects the lipid metabolic disruptions associated with AF pathogenesis.

Currently, few clinical investigations have thoroughly explored the relationship between the TyG index and AF, and the predictive capability of AIP in relation to AF risk remains largely unexplored. Therefore, this study aimed to clarify these

associations, comparatively evaluate their predictive performances, and provide clinical insights into the assessment of AF risk.

## Materials and methods

### Study population

In this retrospective analysis, we reviewed the clinical records of 1,480 patients admitted to the First Hospital of Jilin University between January 1, 2023, and February 29, 2024. After confirming the diagnosis and exclusion criteria, the data of 1,122 patients were finally included, as shown in the flowchart in Fig 1. Persistent AF (PeAF) was identified in 404 patients, and paroxysmal AF (PAF) was identified in 409 patients. Additionally, 309 randomly chosen age-matched inpatients without AF from the same department within the specified timeframe comprised the control group. The data were accessed for research purposes on 01/04/2024. Throughout the data collection process, all participants' information was anonymized. The exclusion criteria were as follows: (1) valvular heart disease of rheumatic origin; (2) cardiomyopathy; (3) congenital cardiac anomalies; (4) chronic pulmonary heart conditions; (5) prior mechanical or biological valve replacement; (6) hyperthyroidism; (7) familial hypertriglyceridemia. Individuals with incomplete lipid profiles or impaired renal function were also excluded. Ethical approval (No. 2024−527) was obtained from the Ethics Committee of the First Hospital of Jilin University. Owing to the retrospective design of this study, the requirement for informed consent was waived.

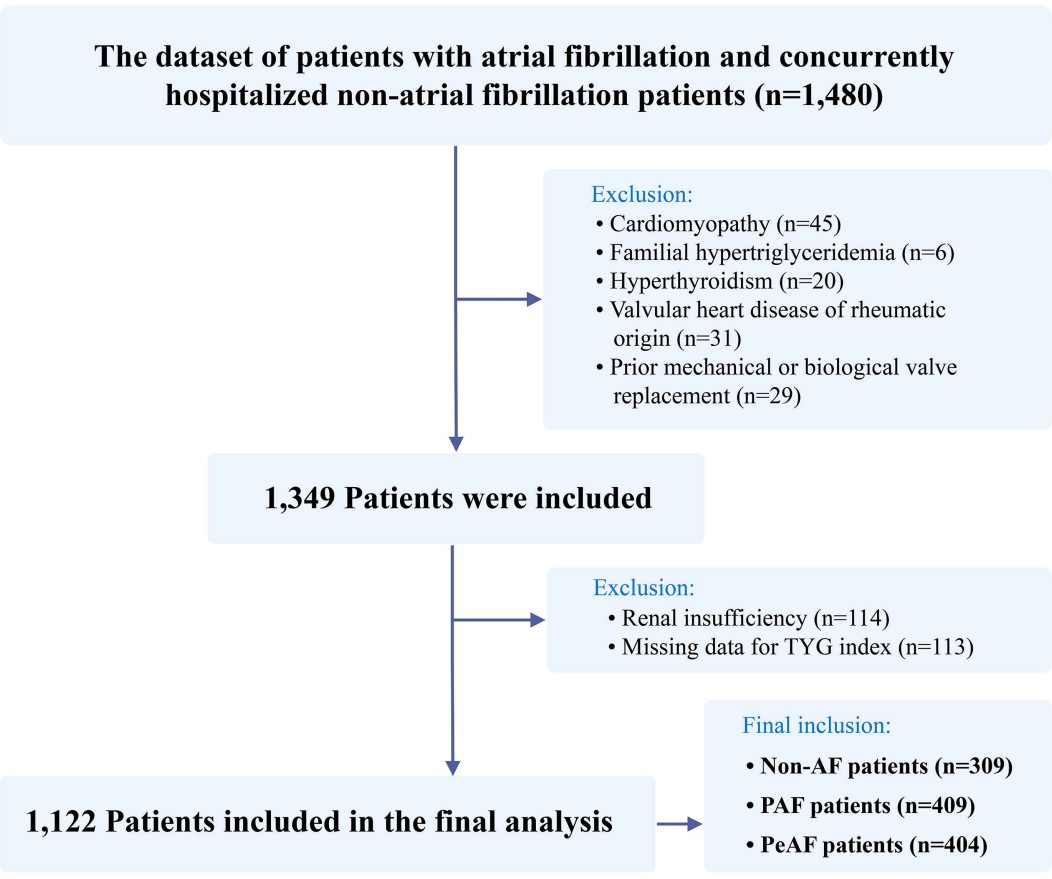

**Fig 1. Study population selection flowchart.**

## Clinical data and definitions

Demographic characteristics and clinical parameters were systematically compiled from inpatient electronic records and organized in a predesigned Excel database (S1 Table). The obtained variables included demographic factors (age and sex), height, weight, smoking, drinking, hypertension, DM, coronary heart disease (CHD), cerebrovascular incidents, and AF subtype. Laboratory measurements: Fasting peripheral venous blood samples were collected from patients the following morning for tests, including alanine aminotransferase (ALT), aspartate aminotransferase (AST), total cholesterol (TC), TG, HDL-C, low-density lipoprotein cholesterol (LDL-C), serum creatinine (SCr), and fasting blood glucose (FBG). All patients underwent an electrocardiogram and a cardiac ultrasound. The cardiac ultrasound primarily measured left ventricular ejection fraction (LVEF), left ventricular end-diastolic dimension (LVEDD), left atrial diameter (LAD), and left ventricular diastolic dysfunction (LVDD). Patients with AF were also required to undergo Holter monitoring.

AF subtypes were classified according to established clinical guidelines: PAF included AF episodes lasting ≤ 7 days, whereas PeAF included AF episodes lasting > 7 days [1]. A trained cardiac ultrasound technician assessed the LVDD. The calculation formulas used were as follows: TyG index = Ln [TG (mg/dL) × FBG (mg/dL)/ 2]; TyG-body mass index (BMI) = TyG index × BMI [17]; and AIP = log [TG (mg/dL)/ HDL-C (mg/dL)] [13].

## Statistical analysis

Statistical analyses were performed using R version 4.3.0 (R Foundation for Statistical Computing) and IBM SPSS Statistics for Windows version 27.0 (IBM Corp., Armonk, N.Y., USA). Data following a normal distribution are presented as mean ± standard deviation, whereas non-normally distributed data are presented as quartiles. Categorical data are presented as n (%) values. A multivariate analysis was conducted to examine the association between AIP, the TyG index, and AF risk. Furthermore, fully adjusted logistic regression models incorporating restricted cubic spline (RCS) analysis were used to examine the non-linear relationships between AIP, TyG index, and AF, including clinical subtypes. Subgroup analyses were stratified according to the specific patient characteristics. Since both TyG index and AIP data do not follow a normal distribution, we performed a Spearman rank correlation analysis to assess the correlation between the two variables. Mediation analysis was conducted with AIP as the exposure variable and TyG as the mediator, or TyG as the exposure variable and AIP as the mediator, with AF as the outcome variable. Receiver operating characteristic (ROC) curves were constructed to evaluate the predictive accuracy of the TyG index, AIP, and their combination for detecting AF; and DeLong's test was used to compare the areas under the curve (AUCs). Sensitivity and specificity were calculated at various cutoff points, and the Youden index was used to identify the optimal cutoff values. Two-sided tests were performed, and $P < 0.05$ was considered statistically significant.

## Results

### Comparison of baseline characteristics

This study included 1,122 participants. The demographic data and clinical parameters of patients with AF (with either paroxysmal or persistent forms) and the non-AF control group are presented in Table 1. Compared with controls, the patients with AF exhibited notably higher prevalence rates of hypertension, DM, CHD, and substantially lower HDL-C levels. Furthermore, the AF cohort exhibited significantly higher TyG index, TyG-BMI index, and AIP. Compared to the non-AF group, the AF group had higher values for BMI, ALT, AST, SCr, uric acid, LAD, LVDD, and LVEDD, with the PeAF group showing even higher values than the PAF group. Additionally, patients with PeAF exhibited a decreased LVEF. Conversely, no significant intergroup differences were observed in TC, LDL-C, or with alcohol consumption.

**Table 1. Comparison of baseline characteristics of the study population.**

| Variables | Total | Non-AF | Paroxysmal AF | Persistent AF | P |
|---|---|---|---|---|---|
| | (n = 1122) | (n = 309) | (n = 409) | (n = 404) | |
| Age, years | 61.00 (55.00, 68.00) | 61.00 (55.00, 69.00) | 61.00 (56.00, 68.00) | 62.00 (55.75, 68.00) | 0.930 |
| Male, n (%) | 634 (56.51) | 137 (44.34) | 224 (54.77) | 273 (67.57) | < 0.001 |
| Smoking, n (%) | 92 (8.20) | 34 (11.00) | 23 (5.62) | 35 (8.66) | 0.031 |
| Drinking, n (%) | 93 (8.29) | 24 (7.77) | 26 (6.36) | 43 (10.64) | 0.079 |
| Hypertension, n (%) | 545 (48.57) | 116 (37.54) | 226 (55.26) | 203 (50.25) | < 0.001 |
| DM, n (%) | 213 (18.98) | 35 (11.33) | 103 (25.18) | 75 (18.56) | < 0.001 |
| CHD, n (%) | 232 (20.68) | 54 (17.48) | 106 (25.92) | 72 (17.82) | 0.005 |
| Stroke, n (%) | 113 (10.07) | 22 (7.12) | 32 (7.82) | 59 (14.60) | < 0.001 |
| BMI, kg/m² | 25.64 (23.38, 27.80) | 24.44 (22.10, 27.01) | 25.88 (23.69, 27.92) | 26.09 (23.67, 28.21) | < 0.001 |
| TC, mmol/L | 4.40 (3.75, 5.13) | 4.45 (3.78, 5.18) | 4.40 (3.78, 5.13) | 4.36 (3.68, 5.04) | 0.406 |
| TG, mmol/L | 1.42 (1.07, 1.98) | 1.11 (0.83, 1.64) | 1.68 (1.34, 2.29) | 1.36 (1.04, 1.92) | < 0.001 |
| HDL-C, mmol/L | 1.05 (0.90, 1.21) | 1.11 (0.96, 1.30) | 1.02 (0.88, 1.14) | 1.04 (0.90, 1.21) | < 0.001 |
| LDL-C, mmol/L | 2.75 (2.28, 3.31) | 2.84 (2.28, 3.36) | 2.73 (2.30, 3.29) | 2.71 (2.23, 3.29) | 0.201 |
| ALT, U/L | 19.90 (14.60, 28.78) | 16.90 (12.70, 24.30) | 20.30 (15.20, 30.60) | 21.35 (15.60, 30.22) | < 0.001 |
| AST, U/L | 20.80 (17.00, 25.60) | 19.80 (16.30, 23.50) | 20.50 (16.70, 25.50) | 21.90 (18.30, 28.45) | < 0.001 |
| SCr, umol/L | 69.30 (59.52, 80.07) | 64.00 (56.60, 74.40) | 68.50 (59.20, 80.00) | 73.05 (62.48, 82.70) | < 0.001 |
| Uric acid,umol/L | 350 (295, 415.75) | 320 (273, 366) | 351(293, 420) | 377.50 (310, 446.25) | < 0.001 |
| FBG, mmol/L | 5.38 (4.91, 6.12) | 5.20 (4.79, 5.60) | 5.58 (5.06, 6.57) | 5.39 (4.92, 6.17) | < 0.001 |
| TyG index | 8.74 (8.45, 9.16) | 8.45 (8.03, 8.86) | 8.97 (8.67, 9.35) | 8.70 (8.40, 9.13) | < 0.001 |
| TyG-BMI index | 226.27 (200.12, 250.49) | 205.38 (183.20, 235.31) | 234.09 (214.90, 254.44) | 228.63 (203.40, 252.18) | < 0.001 |
| AIP | 0.31 (−0.03, 0.71) | 0.00 (−0.35, 0.47) | 0.50 (0.24, 0.87) | 0.27 (−0.04, 0.64) | < 0.001 |
| LAD, mm | 38.00 (33.00, 42.00) | 32.00 (30.00, 36.00) | 36.00 (33.00, 40.00) | 42.00 (40.00, 46.00) | < 0.001 |
| LVEF, % | 62.00 (58.00, 64.00) | 63.00 (62.00, 65.00) | 63.00 (60.00, 64.00) | 59.00 (54.00, 63.00) | < 0.001 |
| LVDD, n(%) | 602 (53.70) | 64 (20.78) | 171 (41.81) | 367 (90.84) | < 0.001 |
| LVEDD, mm | 48.00 (45.00, 51.00) | 46.00 (43.00, 49.00) | 48.00 (45.00, 50.00) | 49.50 (46.00, 54.00) | < 0.001 |

AF, atrial fibrillation; AIP, atherogenic index of plasma; BMI, body mass index; ALT, alanine aminotransferase; AST, aspartate aminotransferase; CHD, coronary heart disease; DM, diabetes mellitus; FBG, fasting blood glucose; HDL-C, high-density lipoprotein cholesterol; LAD, left atrial diameter; LDL-C, low-density lipoprotein cholesterol; LVDD, left ventricular diastolic dysfunction; LVEDD, left ventricular end-diastolic diameter; LVEF, left ventricular ejection fraction; SCr, Serum creatinine; TC, total cholesterol; TG, triglyceride; TyG, triglyceride-glucose.

## Logistic regression analysis

Drawing on clinical experience and data from the Arrhythmia Center of Fuwai Hospital [18], the TyG index was categorized into three separate categories for analysis: lower category representing values < 8.67 (Q1), intermediate category encompassing values from 8.67 to 9.37 (Q2), and higher category for values > 9.37 (Q3). AIP was categorized based on tertiles. After adjusting for potential confounding factors (Model 3), the risk of AF in the Q3 group was significantly higher than that in the Q1 group (OR = 3.55, 95% CI: 1.65–7.60, $P$ = 0.001; Table 2). The risk of AF increased progressively with higher TyG index levels. In Model 3, the risk of AF was highest in the T2 group for AIP, being significantly higher than that in the T1 group (OR = 3.56, 95% CI: 2.24–5.66, $P$ < 0.001), while the T3 group also showed a higher risk than the T1 group (OR = 2.21, 95% CI: 1.18–4.12, $P$ = 0.013). Thus, both AIP and TyG index independently serve as predictors of AF risk.

## RCS analysis

Upon applying RCS regression adjusted for confounders specified in Model 3, nonlinear associations with positive trends were identified between AF risk and both the TyG index and AIP (overall and nonlinear $P$-values < 0.001; Fig 2a and b).

**Table 2. Multifactor regression analysis of the TyG index and AIP with atrial fibrillation risk.**

| | Model 1 | | Model 2 | | Model 3 | |
|---|---|---|---|---|---|---|
| | OR (95% CI) | *P* | OR (95% CI) | *P* | OR (95% CI) | *P* |
| TyG index | 2.75 (1.62–4.66) | <0.001 | 2.70 (1.46–5.00) | 0.002 | 2.96 (1.54–5.67) | 0.001 |
| <8.67 | 1.00 (Reference) | | 1.00 (Reference) | | 1.00 (Reference) | |
| 8.67–9.37 | 2.29 (1.55–3.39) | <0.001 | 2.43 (1.51–3.89) | <0.001 | 2.48 (1.52–4.05) | <0.001 |
| >9.37 | 3.79 (2.04–7.05) | <0.001 | 3.44 (1.69–7.02) | <0.001 | 3.55 (1.65–7.60) | 0.001 |
| AIP | 1.66 (1.01–2.75) | 0.048 | 2.38 (1.29–4.38) | 0.005 | 2.41 (1.29–4.53) | 0.006 |
| T1 | 1.00 (Reference) | | 1.00 (Reference) | | 1.00 (Reference) | |
| T2 | 2.40 (1.67–3.44) | <0.001 | 3.37 (2.16–5.28) | <0.001 | 3.56 (2.24–5.66) | <0.001 |
| T3 | 1.52 (0.93–2.47) | 0.006 | 2.21 (1.21–4.03) | 0.009 | 2.21 (1.18–4.12) | 0.013 |

Model 1: Unadjusted.

Model 2: Adjusted for age, sex, and left atrial diameter.

Model 3: Adjusted for age, sex, smoking, hypertension, diabetes mellitus, coronary heart disease, left atrial diameter, left ventricular ejection fraction, left ventricular end-diastolic diameter, left ventricular diastolic dysfunction, body mass index, and serum creatinine level.

AIP, atherogenic index of plasma; CI, confidence interval; OR, odds ratio; TyG, triglyceride-glucose.

Similar nonlinear patterns were observed in the PAF subgroup (Fig 2c and d). Additionally, a positive nonlinear relationship was observed between the TyG index, AIP, and PeAF risk in the corresponding subgroup (Fig 2e and f).

## Subgroup analysis

For subgroup evaluations, the TyG index was categorized into two groups using the median cutoff, whereas AIP was assessed as a continuous measure. Subgroup analyses were conducted from both categorical and continuous perspectives, along with other clinical parameters, including sex, age, hypertension status, DM, CHD, and LVDD. As shown in Fig 3, the TyG index was significantly associated with AF (OR = 3.03, 95% CI: 2.29–4.00, *P*<0.001), with the highest AF risk observed among individuals with DM (OR = 5.01, 95% CI: 2.34–10.73, *P*<0.001). Similarly, AIP (Fig 4) was significantly associated with AF (OR = 3.79, 95% CI: 2.88–4.99, *P*<0.001), and higher AF risk was noted among participants aged ≥65 years, as well as those with DM or CHD. No significant interactions were observed in the subgroup analyses.

## Correlation analysis

The correlation between the AIP and TyG indices was further investigated. The Spearman correlation method was applied due to the non-normal distribution of both indices. As illustrated in Fig 5, the Spearman correlation revealed a strong positive association, with a coefficient ($\rho$) of 0.8531 (*P*<0.001).

## Mediation analysis

To explore the possible reciprocal mediating effects between the AIP and TyG indices on AF occurrence, AIP and TyG served alternately as exposure and mediator variables. With AF as the outcome variable, a mediation analysis adjusted for covariates from Model 3 was conducted. As demonstrated in Fig 6, the mediation proportions of AIP and TyG in the occurrence of AF were 39.20% (*P*=0.180) and 52.08% (*P*=0.132), respectively. No significant mediating effect of AIP and TyG in the occurrence of AF was observed.

## ROC curve results

We conducted a comparative analysis of AIP, TyG, and their combination in the diagnosis of AF (Table 3). As shown in the ROC curve (Fig 7) analysis, the AUC values for TyG, AIP, and their combination were 0.717, 0.691, and 0.713,

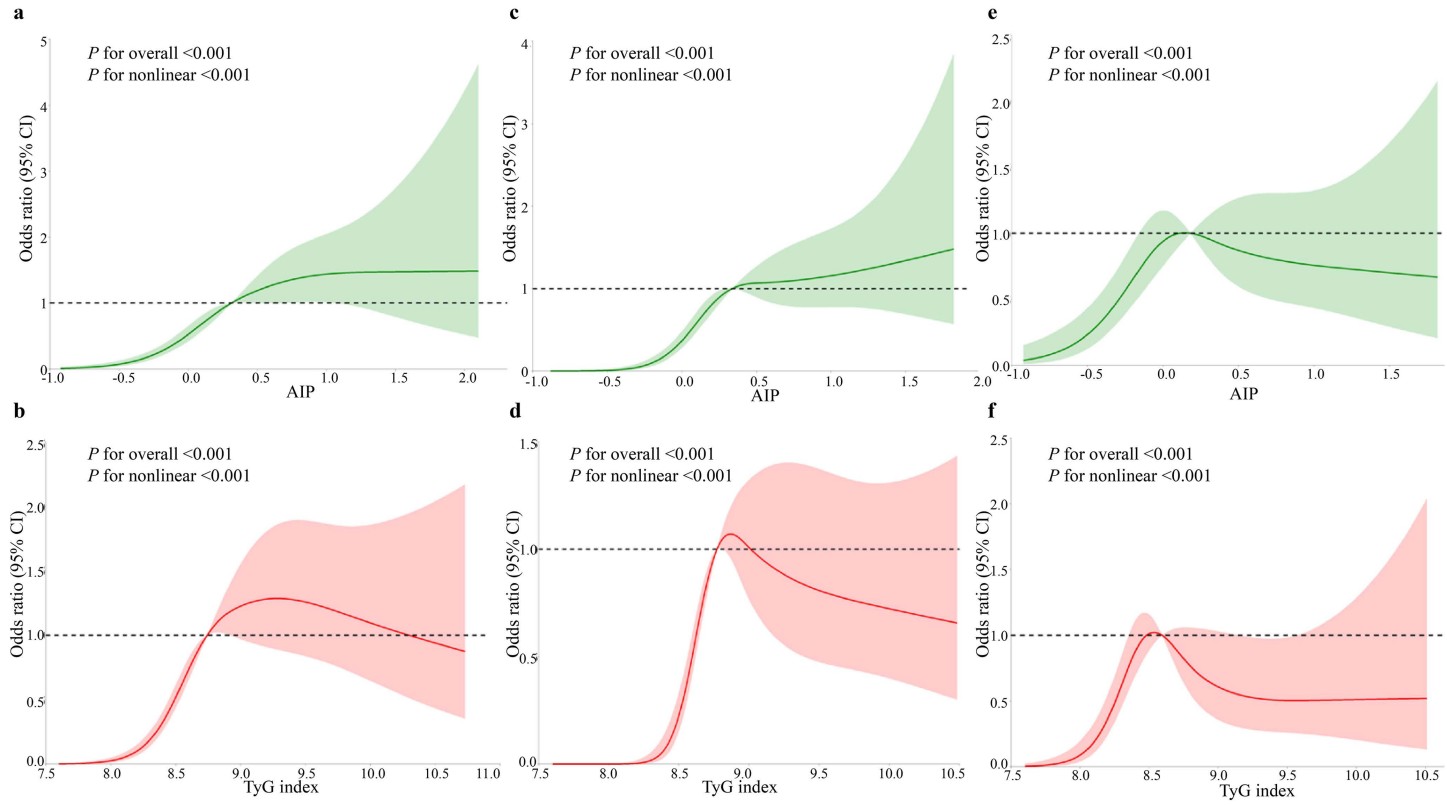

**Fig 2. Association between AIP and TyG index with atrial fibrillation and its subgroups (Adjusted Model 3).**

| Subgroup | n (%) | TyG < Median | TyG ≥ Median | OR (95%CI) | | P | P for interaction |
|---|---|---|---|---|---|---|---|
| | | *No. of events/ No. of total* | | | | | |
| All patients | 1122 (100.00) | 347/561 | 466/561 | 3.03 (2.29 – 4.00) | | < 0.001 | |
| Age | | | | | | | 0.770 |
| <65 | 673 (59.98) | 190/316 | 292/357 | 2.98 (2.10 – 4.23) | | < 0.001 | |
| ≥65 | 449 (40.02) | 157/245 | 174/204 | 3.25 (2.04 – 5.19) | | < 0.001 | |
| Sex | | | | | | | 0.684 |
| Male | 634 (56.51) | 218/313 | 279/321 | 2.89 (1.93 – 4.34) | | < 0.001 | |
| Female | 488 (43.49) | 129/248 | 187/240 | 3.25 (2.20 – 4.83) | | < 0.001 | |
| Hypertension | | | | | | | 0.728 |
| Yes | 545 (48.57) | 177/257 | 252/288 | 3.16 (2.04 – 4.90) | | < 0.001 | |
| No | 577 (51.43) | 170/304 | 214/273 | 2.86 (1.98 – 4.12) | | < 0.001 | |
| DM | | | | | | | 0.108 |
| Yes | 213 (18.98) | 41/62 | 137/151 | 5.01 (2.34 – 10.73) | | < 0.001 | |
| No | 909 (81.02) | 306/499 | 329/410 | 2.56 (1.89 – 3.47) | | < 0.001 | |
| CHD | | | | | | | 0.715 |
| Yes | 232 (20.68) | 81/121 | 97/111 | 3.42 (1.74 – 6.73) | | < 0.001 | |
| No | 890 (79.32) | 266/440 | 369/450 | 2.98 (2.19 – 4.05) | | < 0.001 | |
| LVDD | | | | | | | 0.808 |
| Yes | 602 (53.65) | 249/297 | 289/305 | 3.48 (1.93 – 6.29) | | < 0.001 | |
| No | 520 (46.35) | 98/264 | 177/256 | 3.80 (2.64 – 5.46) | | < 0.001 | |

1  5.9  10.8

**Fig 3. Association between the TyG index and AF risk.**

| Subgroup | n (%) | OR (95%CI) | | P | P for interaction |
|---|---|---|---|---|---|
| All patients | 1,122 (100.00) | 3.79 (2.88 – 4.99) | | < 0.001 | |
| Age | | | | | 0.252 |
| <65 | 673 (59.98) | 3.44 (2.48 – 4.79) | | < 0.001 | |
| ≥65 | 449 (40.02) | 4.86 (2.97 – 7.94) | | < 0.001 | |
| Sex | | | | | 0.749 |
| Male | 634 (56.51) | 3.49 (2.36 – 5.15) | | < 0.001 | |
| Female | 488 (43.49) | 3.82 (2.58 – 5.66) | | < 0.001 | |
| Hypertension | | | | | 0.934 |
| Yes | 545 (48.57) | 3.86 (2.47 – 6.04) | | < 0.001 | |
| No | 577 (51.43) | 3.77 (2.66 – 5.34) | | < 0.001 | |
| DM | | | | | 0.551 |
| Yes | 213 (18.98) | 2.93 (1.35 – 6.34) | | 0.006 | |
| No | 909 (81.02) | 3.78 (2.81 – 5.07) | | < 0.001 | |
| CHD | | | | | 0.884 |
| Yes | 232 (20.68) | 3.96 (2.02 – 7.79) | | < 0.001 | |
| No | 890 (79.32) | 3.75 (2.78 – 5.07) | | < 0.001 | |
| LVDD | | | | | 0.541 |
| Yes | 602 (53.65) | 4.49 (2.59 – 7.78) | | < 0.001 | |
| No | 520 (46.35) | 5.55 (3.73 – 8.25) | | < 0.001 | |

**Fig 4. Association between AIP and AF risk.**

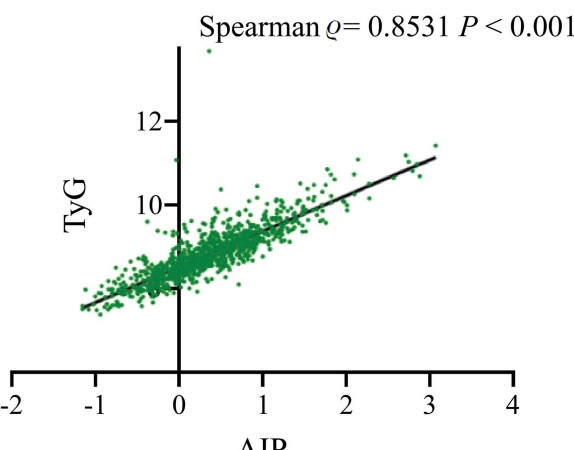

**Fig 5. Correlation analysis between the AIP and TyG index.**

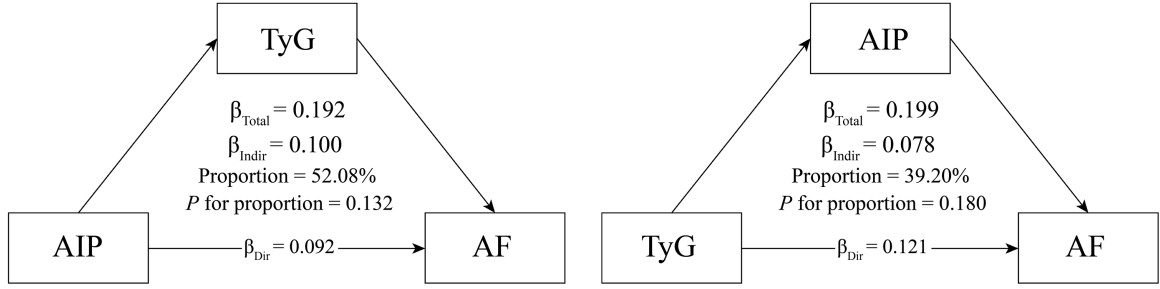

**Fig 6. Mediation analysis of the relationship between the AIP and TyG index with AF.**

**Table 3. Comparison of the ability of AIP, TyG, and their combination to predict atrial fibrillation risk.**

|  | AUC | Sensitivity | Specificity | Youden | Cutoff |
|---|---|---|---|---|---|
| AIP | 0.691 | 0.828 | 0.498 | 0.326 | −0.008 |
| TyG | 0.717 | 0.856 | 0.511 | 0.367 | 8.448 |
| AIP+TyG | 0.713 | 0.822 | 0.537 | 0.359 | 0.651 |

AIP, atherogenic index of plasma; AUC, area under the curve; TyG, triglyceride-glucose

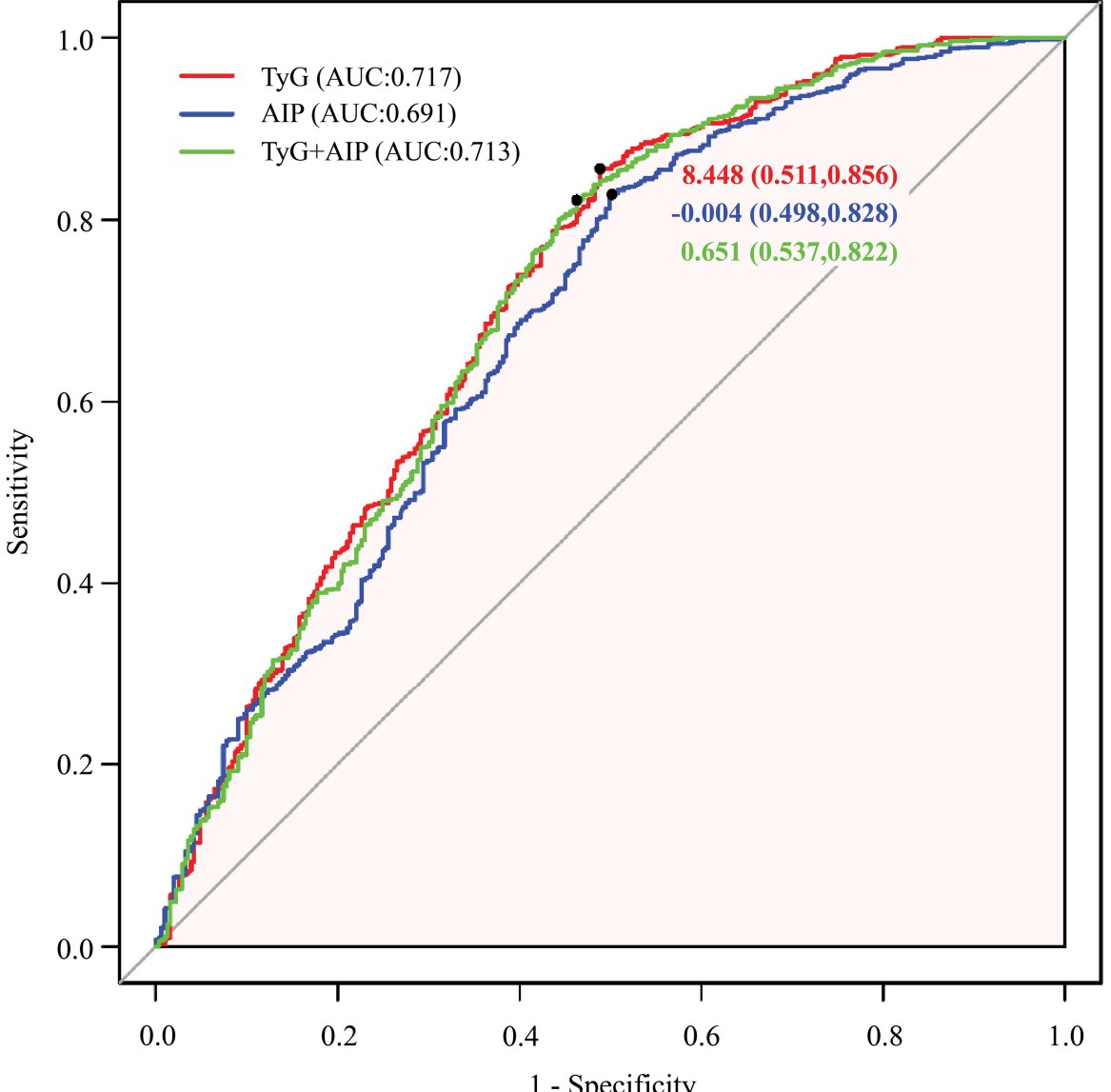

**Fig 7. ROC curves evaluating AF risk using the TyG index and AIP, in combination and individually.**

respectively. DeLong's test indicated that the differences were not statistically significant ($P = 0.3205$), suggesting that TyG and AIP exhibited similar predictive efficacy for AF risk.

## Discussion

This study represents an initial attempt to explore the relationship between AIP and AF while concurrently comparing its predictive effectiveness with that of the TyG index. After adjusting for multiple covariates, the TyG index and AIP were significantly positively correlated with AF, indicating that TyG and AIP are independent predictors of AF. Furthermore, RCS analyses revealed significant positive nonlinear relationships between both indices and AF. In the subgroup analyses, both AIP and TyG were significantly positively correlated with AF in different subgroups. In the TyG subgroups, patients with DM had the highest risk of AF; in the AIP subgroups, patients aged ≥ 65 years, DM, and CHD had a higher risk of AF, with no significant interactions. Correlation analysis revealed a significant association between the TyG index and AIP. However, mediation analysis revealed no reciprocal mediation effect between the TyG index and AIP in AF occurrence. ROC curve analysis showed that AIP and TyG had comparable predictive abilities.

IR, characterized by diminished insulin responsiveness in peripheral tissues, is central to metabolic syndrome and is intricately associated with disorders such as type 2 DM and numerous cardiovascular diseases. Recently, IR has been recognized as a standalone prognostic indicator of cardiovascular events [19]. A comprehensive analysis revealed that even when accounting for traditional cardiovascular risk factors, IR maintained its notable predictive significance for an increased risk of heart failure [20]. An animal study showed that in IR animals, the heart is in a state of IR, making it highly susceptible to AF and other atrial tachyarrhythmias. Compared to the control group, IR animals exhibited significantly increased expression of active transforming growth factor β 1 and matrix metalloproteinase-9 in the atria, indicating a correlation between the upregulation of these pro-fibrotic markers and AF occurrence [21]. Similarly, studies examining non-diabetic groups in Asia confirmed that IR is a distinct risk factor for AF, surpassing conventional cardiovascular risk elements [22].

An increasing body of literature highlights the robust association between the TyG index and IR, emphasizing its comparable or superior predictive power to that of traditional IR measurement approaches. Compared with these established methodologies, the TyG index has equal or greater accuracy in identifying IR, offering enhanced sensitivity and specificity [23–28]. Clinically, the TyG index has several advantages over traditional IR evaluation methods, including convenience, practicality, and cost-effectiveness. Mounting evidence indicates that elevated TyG index values are positively associated with a greater likelihood of developing AF, implying that higher TyG index values predict increased AF risk [29–31]. Additionally, the TyG index has demonstrated significant potential for predicting AF in certain patient populations [30]. For example, a retrospective evaluation of patients who underwent ventricular septal myectomy for hypertrophic obstructive cardiomyopathy showed that elevated TyG indices independently indicated a higher postoperative AF risk, with moderate predictive effectiveness [32]. Another retrospective study showed that in patients with acute coronary syndrome, the TyG index is an independent predictor of AF [33].

Furthermore, data from the UK Biobank cohort reported a nonlinear U-shaped curve characterizing the relationship between AF incidence and the TyG index [12]. This observation closely mirrors the outcomes of an American community-based study involving women without a prior cardiovascular history [34]. A recent meta-analysis showed that the TyG index is associated with the risk of AF, including AF recurrence after ablation, postoperative AF, and lone AF [35]. The findings of our study are consistent with those of previous studies, multivariate logistic regression, and RCS analyses showing that the TyG index is positively associated with the risk of AF and exhibits a nonlinear relationship. Our study supports the use of the TyG index alone for predicting the occurrence of AF. TyG can serve as a simple and cost-effective clinical tool for quantifying IR [36,37]. It may also function as an early screening tool for AF, particularly in high-risk populations. By monitoring the TyG index, abnormalities such as IR can be identified promptly, thereby providing a basis for early intervention and risk management [38]. However, as TyG is an indirect indicator of IR, its value may be influenced by

factors such as fasting status and lipid metabolism abnormalities. Therefore, its predictive value has certain limitations and cannot fully replace a comprehensive metabolic assessment.

AIP is a highly effective biomarker for assessing atherosclerosis. A significant relationship between higher AIP values and stroke incidence has been documented in middle-aged and older populations using data from the CHARLS database, with this relationship notably varying according to glucose metabolic status [39]. Previous studies have highlighted the robust association between AIP and cardiovascular conditions, particularly in patients with diabetes [40]. In recent years, many proteomic and metabolomic studies have found that dyslipidemia is also involved in the occurrence and development of AF [41,42]. In an experimental study, AF induced by Angiotensin II, along with left atrial enlargement, fibrosis, inflammatory responses, and oxidative stress, was markedly enhanced in mice lacking the adipose triglyceride lipase (*Atgl*) gene compared with wild-type mice. ATGL primarily catalyzes TG hydrolysis. ATGL overexpression can improve cardiac energy metabolism, contractile function, and cardiac remodeling, suggesting a correlation between TG and AF [43]. HDL-C has recently been confirmed to reverse cardiac remodeling by alleviating myocardial hypertrophy and fibrosis [44]. Several studies have consistently identified HDL-C and TG levels as lipid parameters associated with AF incidence [45–47]. Seong-Min Kim's study showed that, compared to patients with paroxysmal supraventricular tachycardia, the AF group exhibits elevated TG and C-reactive protein levels, along with hyperuricemia [15]. A recent study showed that low HDL-C levels and elevated AIP are significantly associated with left atrial enlargement [48]. Similarly, a separate case-control investigation observed significantly higher TG levels and markedly reduced HDL-C concentrations in patients with AF than in matched controls [49], consistent with our findings. In addition, we found no statistically significant differences in LDL-C and TC between the two groups. AIP, which considers changes in both HDL-C and TG levels, provides a more comprehensive reflection of lipid metabolism [50]. Therefore, we conducted logistic regression and RCS analysis, which showed that AIP was significantly associated with AF in a nonlinear relationship. In the subgroup analysis, AF risk was higher in the AIP group among those aged ≥ 65 years, with DM and CHD, differing from the TyG group, where AF risk was highest in patients with DM. This suggests that TyG may primarily affect AF through the IR pathway, whereas AIP influences AF through lipid metabolism. Additionally, our study results show that AIP is significantly correlated with TyG and, compared with the widely used TyG index, a surrogate marker for IR, demonstrates a comparable ability to assess AF risk. Thus, the application of IR is consistent with that of AIP in predicting AF. This may be attributed to the overlap between the two indices in their pathological pathways (e.g., left atrial enlargement/oxidative stress, inflammation/atrial remodeling). Our results showed the mediation analysis was negative; therefore, we consider them to be more complementary than interchangeable clinical tools. We recommend their combined use with traditional risk factors for improved risk stratification, while emphasizing the need for prospective validation in diverse populations.

This study has some limitations. First, because it was retrospective, controlling for all potential confounders was challenging. Specifically, the higher prevalence of hypertension, DM, and CHD in patients with PAF than in those with PeAF may have introduced some bias when comparing the TyG index and AIP between these AF subtypes. Second, the sample size was small, and the single-center setting constrained the generalizability of the results. Future studies with larger multicenter cohorts are warranted. Third, this study used only baseline TyG and AIP values measured at hospital admission without exploring temporal changes or the therapeutic impact on clinical outcomes. Finally, the retrospective design may have constrained definitive causal interpretations between the two indices and AF risk. Therefore, future prospective multicenter studies are essential to validate and expand our findings.

## Conclusions

Elevated AIP and TyG index levels were significantly and nonlinearly associated with an increased AF risk. Furthermore, the predictive capabilities of AIP for AF appeared to be comparable to those of the TyG index. Subsequent studies should further evaluate the clinical utility of AIP and the TyG index and integrate these indices with other cardiovascular biomarkers to develop a more comprehensive clinical strategy for AF risk stratification.

## Supporting information

**S1 Table. Patient clinical data set.**
(XLSX)

## Acknowledgments

We express our sincere gratitude to all the participants involved in this study.

## Author contributions

**Data curation:** Xuebo Li.

**Formal analysis:** Shangzhi Shu.

**Investigation:** Shangzhi Shu.

**Methodology:** Xuebo Li, Shangzhi Shu.

**Software:** Xuebo Li, Shangzhi Shu, Chaoqun Huang.

**Supervision:** Shuyan Li.

**Validation:** Xuebo Li, Shangzhi Shu.

**Writing – original draft:** Xuebo Li.

**Writing – review & editing:** Shuyan Li.

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
