## [Decision Letter · Decision Letter 0]

20 Oct 2025

Dear Dr. Li,

Thank you for submitting your manuscript to PLOS ONE. After careful consideration, we feel that it has merit but does not fully meet PLOS ONE’s publication criteria as it currently stands. Therefore, we invite you to submit a revised version of the manuscript that addresses the points raised during the review process.

**ACADEMIC EDITOR:  Major revision**

We look forward to receiving your revised manuscript.

Kind regards,

Marwan Salih Al-Nimer, MD, PhD

Academic Editor

PLOS ONE

Journal Requirements:

2. We note that there is identifying data in the Supporting Information file < S1_Table.xlsx>. Due to the inclusion of these potentially identifying data, we have removed this file from your file inventory. Prior to sharing human research participant data, authors should consult with an ethics committee to ensure data are shared in accordance with participant consent and all applicable local laws.

-Location data

Please remove or anonymize all personal information (<Age>), ensure that the data shared are in accordance with participant consent, and re-upload a fully anonymized data set. Please note that spreadsheet columns with personal information must be removed and not hidden as all hidden columns will appear in the published file.

**Additional Editor Comments:**

This study needs the following points to be clarified

1: The title does not reflect the objectives and the conclusion remarks

2: Line 51: support this paragraph with many references with explanation.

3: Line 103: this study is retrospective. Can you explain how the procedures of the laboratory measurements were done?

4: Line 134: The number of included participants was 1,122 while the excluding criteria included so many causes. Therefore explain in details the participants of the study. The flowchart is not informative

5: Line 140: What about the data of liver enzymes, TC....etc.

6: P-values should be of 3 digit instead of p< or p> 0.05

7: In statistical analysis: clearly mention the type of bivariate correlation, the methods of mediation analysis, and the ROC ( sensitivity, specificity, Youden index, cutoff values...etc). and these parameters should be clearly presented in the tables and figures

Reviewers' comments:

Reviewer's Responses to Questions

**Comments to the Author**

1. Is the manuscript technically sound, and do the data support the conclusions?

Reviewer #1: Yes

Reviewer #2: Yes

2. Has the statistical analysis been performed appropriately and rigorously?

Reviewer #1: I Don't Know

Reviewer #2: I Don't Know

3. Have the authors made all data underlying the findings in their manuscript fully available?

Reviewer #1: Yes

Reviewer #2: Yes

4. Is the manuscript presented in an intelligible fashion and written in standard English?

Reviewer #1: Yes

Reviewer #2: Yes

Reviewer #1: This is a retrospective study to understand the correlation of TyG and AIP with AF. With obesity being one of the strongest risk factor for Atrial fibrillation, this study helps to add new data and in this realm and directs for need of further research in the area. The manuscript is well written. Limitations of the study are documented.

Reviewer #2: Well done

Your statistical analyses were robust but it should be broken down in clearly defined terms while discussing the results.

i believe it should be replicable, in order to increase the power of this study.

i got confused and couldn't really appreciate the interpretation of your robust results in the discussion. a revision in use of simpler words that truly reflects the results should be use in the discussion and more importantly make references to similar studies already published, this would drive home your points clearly.

can you write on / highlight implications / usefullness of this study in clinical practice? For Example; Does this study supports the use TyG alone to predict AF? if so what are the flaws ? Is the use of IR same as the AIP in predicting AF in clinical practice? and explain reasons based on your results..vis a viz previous publications.

**Do you want your identity to be public for this peer review?** For information about this choice, including consent withdrawal, please see our Privacy Policy

Reviewer #1: No

Reviewer #2: No

---

## [Author Response · Author response to Decision Letter 1]

27 Nov 2025

Dear Editor and Reviewers,

We would like to express our sincere gratitude for the opportunity to revise our manuscript titled: “Association of atherogenic index of plasma and triglyceride-glucose index with atrial fibrillation: a retrospective study” (Manuscript ID: PONE-D-25-48225). We sincerely appreciate the insightful and constructive feedback from you and the reviewers, which has been invaluable in improving the overall quality of our manuscript. We have carefully reviewed and addressed all the comments and suggestions, making the necessary revisions accordingly. A detailed, point-by-point response to each comment is provided below.

Response to Editor’s Comments:

Editor Comment 1: The title does not reflect the objectives and the conclusion remarks

Response: Thank you for your comment. The title has been revised to “Association of atherogenic index of plasma and triglyceride-glucose index with atrial fibrillation: a retrospective study.”

Editor Comment 2: Line 51: support this paragraph with many references with explanation.

Response: Thank you for your valuable suggestion, which has helped improve the quality of our manuscript. We have revised the text as follows:

“Insulin resistance (IR) significantly influences AF pathogenesis through diverse mechanisms [2]. First, IR promotes the onset of AF through structural and electrical remodeling, including left atrial enlargement, fibrosis, and intracellular calcium homeostasis imbalance [3,4]. Oxidative stress and inflammation caused by IR lead to atrial remodeling through a synergistic interaction involving mechanisms such as reactive oxygen species (ROS) damage, ion channel dysfunction, and the release of inflammatory factors [5-8]. In addition, IR can lead to autonomic nervous system disturbances, such as decreased vagal activity and dysregulated autonomic balance [9,10].” (lines 51–58).

Editor Comment 3: Line 103: this study is retrospective. Can you explain how the procedures of the laboratory measurements were done?

Response: Thank you for your question. We have made corrections regarding the laboratory measurements, and the text has been revised as follows:

“Laboratory measurements: Fasting peripheral venous blood samples were collected from patients the following morning for tests, including alanine aminotransferase (ALT), aspartate aminotransferase (AST), total cholesterol (TC), TG, HDL-C, low-density lipoprotein cholesterol (LDL-C), serum creatinine (SCr), and fasting blood glucose (FBG). All patients underwent an electrocardiogram and a cardiac ultrasound. The cardiac ultrasound primarily measured left ventricular ejection fraction (LVEF), left ventricular end-diastolic dimension (LVEDD), left atrial diameter (LAD), and left ventricular diastolic dysfunction (LVDD). Patients with AF were also required to undergo Holter monitoring.” (lines 105–112).

Editor Comment 4: The number of included participants was 1,122 while the excluding criteria included so many causes. Therefore explain in details the participants of the study. The flowchart is not informative

Response: Thank you for your valuable feedback. The text has been revised as follows:

“In this retrospective analysis, we reviewed the clinical records of 1,480 patients admitted to the First Hospital of Jilin University between January 1, 2023, and February 29, 2024. After confirming the diagnosis and exclusion criteria, the data of 1,122 patients were finally included…” (lines 82–85)

We have also redrawn the flowchart detailing the participants of the study. The specific exclusion criteria and number of patients excluded for each criterion are detailed in Fig. 1.

Fig 1. Study population selection flowchart.

Editor Comment 5: What about the data of liver enzymes, TC....etc.

Response: Thank you for your comment. We have added these data, and the text has been revised as follows:

“Furthermore, the AF cohort exhibited significantly higher TyG index, TyG-BMI index, and AIP. Compared to the non-AF group, the AF group had higher values for BMI, ALT, AST, SCr, uric acid, LAD, LVDD, and LVEDD, with the PeAF group showing even higher values than the PAF group. Additionally, patients with PeAF exhibited a decreased LVEF. Conversely, no significant intergroup differences were observed in TC, LDL-C, or with alcohol consumption.” (lines 145–150).

Editor Comment 6: P-values should be of 3 digit instead of p< or p> 0.05

Response: We sincerely appreciate your feedback. We have corrected the P values throughout the manuscript as suggested.

Editor Comment 7: In statistical analysis: clearly mention the type of bivariate correlation, the methods of mediation analysis, and the ROC (sensitivity, specificity, Youden index, cutoff values...etc). and these parameters should be clearly presented in the tables and figures

Response: Thank you for your comment. We have revised the description of the type of correlation analysis and the methods of mediation analysis. The ROC analysis in the statistical analysis section has been updated as follows:

“Since both TyG index and AIP data do not follow a normal distribution, we performed a Spearman rank correlation analysis to assess the correlation between the two variables. Mediation analysis was conducted with AIP as the exposure variable and TyG as the mediator, or TyG as the exposure variable and AIP as the mediator, with AF as the outcome variable. Receiver operating characteristic (ROC) curves were constructed to evaluate the predictive accuracy of the TyG index, AIP, and their combination for detecting AF; and DeLong’s test was used to compare the areas under the curve (AUCs). Sensitivity and specificity were calculated at various cutoff points, and the Youden index was used to identify the optimal cutoff values.” (lines 128–136)

In addition, we have added Table 3 to provide supplementary information on specific parameters such as sensitivity, specificity, Youden index, and cutoff values (lines 233–235).

Table 3. Comparison of the ability of AIP, TyG, and their combination to predict atrial fibrillation risk

AUC Sensitivity Specificity Youden Cutoff

AIP 0.691 0.828 0.498 0.326 -0.008

TyG 0.717 0.856 0.511 0.367 8.448

AIP + TyG 0.713 0.822 0.537 0.359 0.651

AIP, atherogenic index of plasma; AUC, area under the curve; Tyg, triglyceride-glucose

Response to Reviewer’s Comments:

Reviewer 1’s Comments: This is a retrospective study to understand the correlation of TyG and AIP with AF. With obesity being one of the strongest risk factor for Atrial fibrillation, this study helps to add new data and in this realm and directs for need of further research in the area. Themanuscript is well written. Limitations of the study are documented.

Response: Thank you for your positive and encouraging comments. We are pleased that you found our manuscript well written and recognized its contribution to understanding the relationship between TyG, AIP, and atrial fibrillation. We also appreciate your acknowledgment of the study’s documented limitations. We will continue to refine our research and explore the mechanisms underlying these associations in future studies, particularly in populations with obesity, to provide more comprehensive evidence.

Reviewer 2’s Comments:

Reviewer 2 Comment 1: Your statistical analyses were robust but it should be broken down in clearly defined terms while discussing the results.

Response: Thank you for your valuable suggestions. In response to your comments, we have revised the discussion of the results as follows:

“After adjusting for multiple covariates, the TyG index and AIP were significantly positively correlated with AF, indicating that TyG and AIP are independent predictors of AF. Furthermore, RCS analyses revealed significant positive nonlinear relationships between both indices and AF. In the subgroup analyses, both AIP and TyG were significantly positively correlated with AF in different subgroups. In the TyG subgroups, patients with DM had the highest risk of AF; in the AIP subgroups, patients aged ≥ 65 years, DM, and CHD had a higher risk of AF, with no significant interactions. Correlation analysis revealed a significant association between the TyG index and AIP. However, mediation analysis revealed no reciprocal mediation effect between the TyG index and AIP in AF occurrence. ROC curve analysis showed that AIP and TyG had comparable predictive abilities.” (lines 243–252)

Reviewer 2 Comment 2: i got confused and couldn't really appreciate the interpretation of your robust results in the discussion. a revision in use of simpler words that truly reflects the results should be use in the discussion and more importantly make references to similar studies already published, this would drivehome your points clearly.

Response: Thank you for your valuable feedback on our manuscript. Regarding the discussion section, we have carefully considered your concerns and made revisions. We have simplified the language to present the interpretation of the results more clearly, ensuring that the findings are effectively communicated. Additionally, we have included references to similar studies to better support our points and strengthen the credibility of the discussion.

Reviewer 2 Comment 3: can you write on / highlight implications / usefullness of this study in clinical practice? For Example; Does this study supports the use TyG alone to predict AF? if so what are the flaws ?

Response: Thank you for your valuable suggestions. Our study supports the use of TyG alone to predict atrial fibrillation, although it has certain limitations. In response to your comment, we have added the following text to the Discussion:

“Our study supports the use of the TyG index alone for predicting the occurrence of AF. TyG can serve as a simple and cost-effective clinical tool for quantifying IR [36,37]. It may also function as an early screening tool for AF, particularly in high-risk populations. By monitoring the TyG index, abnormalities such as IR can be identified promptly, thereby providing a basis for early intervention and risk management [38]. However, as TyG is an indirect indicator of IR, its value may be influenced by factors such as fasting status and lipid metabolism abnormalities. Therefore, its predictive value has certain limitations and cannot fully replace a comprehensive metabolic assessment.” (lines 287–295)

Reviewer 2 Comment 4: Is the use of IR same as the AIP in predicting AF in clinical practice? and explain reasons based on yourresults..vis a viz previous publications.

Response: Thank you for your comments. We have incorporated your suggestions into the manuscript. Since there are very few studies on the association between AIP and AF, we have appropriately cited literature on the relationships between AIP components—HDL-C and TG—and AF. In addition, we have cited the study most closely related to our topic, which examined the association between AIP and left atrial enlargement, to provide further supporting explanation. Our research shows that using IR to predict atrial fibrillation is similar to using AIP, which may be due to the significant correlation between AIP and TyG, and the high overlap of their underlying pathological mechanisms in causing atrial fibrillation. The detailed revision is as follows:

“Additionally, our study results show that AIP is significantly correlated with TyG and, compared with the widely used TyG index, a surrogate marker for IR, demonstrates a comparable ability to assess AF risk. Thus, the application of IR is consistent with that of AIP in predicting AF. This may be attributed to the overlap between the two indices in their pathological pathways (e.g., left atrial enlargement /oxidative stress, inflammation/atrial remodeling). Our results showed the mediation analysis was negative; therefore, we consider them to be more complementary than interchangeable clinical tools. We recommend their combined use with traditional risk factors for improved risk stratification, while emphasizing the need for prospective validation in diverse populations.” (lines 324–333)

---

## [Editor Report · Decision Letter 1]

1 Dec 2025

Association of atherogenic index of plasma and triglyceride-glucose index with atrial fibrillation: a retrospective study

PONE-D-25-48225R1

Dear Dr. Shuyan Li,

We’re pleased to inform you that your manuscript has been judged scientifically suitable for publication and will be formally accepted for publication once it meets all outstanding technical requirements.

Kind regards,

Marwan Salih Al-Nimer, MD, PhD

Academic Editor

PLOS ONE
---

## [Editor Report · Acceptance letter]

PONE-D-25-48225R1

PLOS One

Dear Dr. Li,

I'm pleased to inform you that your manuscript has been deemed suitable for publication in PLOS One. Congratulations! Your manuscript is now being handed over to our production team.

Kind regards,

on behalf of

Professor Marwan Salih Al-Nimer

Academic Editor

PLOS One